# SPECT/CT Radiomics for Differentiating between Enchondroma and Grade I Chondrosarcoma

**Hyukjin Yoon** [1] , **Woo Hee Choi** [1] , **Min Wook Joo** [2] , **Seunggyun Ha** [3] **and Yong-An Chung** [4,*]

1   Division of Nuclear Medicine, Department of Radiology, St. Vincent's Hospital, College of Medicine, The Catholic University of Korea, Seoul 06591, Republic of Korea; atznawa@gmail.com (H.Y.); wh0522@catholic.ac.kr (W.H.C.)

2   Department of Orthopedic Surgery, St. Vincent's Hospital, College of Medicine, The Catholic University of Korea, Seoul 06591, Republic of Korea; mwjoo@catholic.ac.kr

3   Division of Nuclear Medicine, Department of Radiology, Seoul St. Mary's Hospital, College of Medicine, The Catholic University of Korea, Seoul 06591, Republic of Korea; seunggyun.ha@gmail.com

4   Division of Nuclear Medicine, Department of Radiology, Incheon St. Mary's Hospital, College of Medicine, The Catholic University of Korea, Seoul 06591, Republic of Korea

*   Correspondence: yongan@catholic.ac.kr

**Abstract:** This study was performed to assess the value of SPECT/CT radiomics parameters in differentiating enchondroma and atypical cartilaginous tumors (ACTs) located in the long bones. Quantitative HDP SPECT/CT data of 49 patients with enchondromas or ACTs in the long bones were retrospectively reviewed. Patients were randomly split into training (*n* = 32) and test (*n* = 17) data, and SPECT/CT radiomics parameters were extracted. In training data, LASSO was employed for feature reduction. Selected parameters were compared with classic quantitative parameters for the prediction of diagnosis. Significant parameters from training data were again tested in the test data. A total of 12 (37.5%) and 6 (35.2%) patients were diagnosed as ACTs in training and test data, respectively. LASSO regression selected two radiomics features, zone-length non-uniformity for zone (ZLNU$_{GLZLM}$) and coarseness for neighborhood grey-level difference (Coarseness$_{NGLDM}$). Multivariate analysis revealed higher ZLNU$_{GLZLM}$ as the only significant independent factor for the prediction of ACTs, with sensitivity and specificity of 85.0% and 58.3%, respectively, with a cut-off value of 191.26. In test data, higher ZLNU$_{GLZLM}$ was again associated with the diagnosis of ACTs, with sensitivity and specificity of 85.0% and 58.3%, respectively. HDP SPECT/CT radiomics may provide added value for differentiating between enchondromas and ACTs.

**Keywords:** HDP bone SPECT/CT; radiomics; enchondromas; chondrosarcoma

## 1. Introduction

Chondrosarcoma and enchondroma are the most common malignant and benign cartilage-forming bone tumors, respectively. Chondrosarcoma is classified as grade 1–3. Grade 1 chondrosarcomas are now referred to as atypical cartilaginous tumors (ACTs) [1]. While ACTs represent a low-grade tumor, surgical treatment is currently regarded as the only form of curative management. Enchondroma is a benign tumor, and a wait-and-see policy is permissible [2]. Due to such different management strategies, proper differentiation between ACTs and enchondromas is crucial. However, ACTs and enchondromas show very similar radiographical and histologic characteristics [3].

Radionuclide bone imaging is used to observe osteoblastic activity associated with bone diseases. Bone scintigraphy has been employed in an attempt to differentiate between enchondromas and chondrosarcomas. Bone scintigraphy can reflect cortical destruction and permeation due to chondroid tumor [4]. Chondrosarcomas generally demonstrate marked heterogeneous radionuclide uptake in only a small proportion of cases. Single photon emission computed tomography (SPECT) provides three-dimensional information

of radiotracer uptake. Recently, technological advancements have allowed combined modality of SPECT/CT to be used to provide a quantitative assessment of radiotracer distribution, similar to PET/CT [5].

Previously, we evaluated the value of quantitative SPECT/CT for differentiating ACTs from enchondromas, and we demonstrated that ACTs show higher SUVmax compared to enchondromas [6]. With a cut-off value of 15.6 for SUVmax, its sensitivity and specificity were 86% and 75% for differentiating between enchondroma and ACTs, respectively.

Texture analysis has been used as a valuable tool for medical images in recent years [7]. Studies have suggested that certain texture parameters that reflect intratumoral heterogeneity can serve as valuable prognostic markers [8]. Texture analysis with MR images have shown the potential for differentiating chondrosarcoma and enchondromas [9,10]. While there have been several attempts to use radiomics from anatomic imaging, the value of SPECT/CT radiomics have not yet been evaluated in bone tumors.

We hypothesized that bone metabolic radiomics parameters would be valuable biomarkers for the diagnosis of ACTs, as they may represent the intratumoral heterogeneity. The purpose of our study was to investigate the usefulness of texture analysis with quantitative SPECT/CT in differentiating enchondromas and ACTs.

## 2. Materials and Methods

### 2.1. Patient Selection

The retrospective study design received approval from the institutional review board (IRB) of St. Vincent's Hospital. Informed consent was waived by the IRB. This study included patients with suspected enchondroma or ACTs originating in the long bones who underwent bone SPECT/CT between December 2015 and June 2021. Patients who had surgical removal or biopsy before SPECT/CT were excluded. Clinical and radiological data, including age, gender, tumor location, and imaging findings, were gathered from the medical records. Patients were randomly distributed into training and test data, in a 2:1 ratio, with a random number generator.

### 2.2. Image Acquisition

All SPECT/CT scans were conducted using an NMCT/670 SPECT/CT scanner (GE Healthcare, Waukesha, WI, USA). First, 800–1100 MBq of Tc-99m hydroxymethylene diphosphonate (HDP) was injected. SPECT/CT images of tumor site were obtained 4 h after radiotracer injection. CT acquisition was done by the following parameters: peak energy at 140 keV with 10% window and step-and-shot mode acquisition (25 s per step and 30 steps per detector) with 6° angular increments. For SPECT image reconstruction, an iterative ordered subset expectation maximization algorithm was employed (four iterations; 10 subsets), with CT-based attenuation correction, scatter correction, and resolution recovery carried out on a Xeleris imaging workstation (version 4.0, GE Healthcare, Waukesha, WI, USA). The reconstructed images had a matrix size of $128 \times 128$ with a section thickness of 4.42 mm. The minimal source-to-collimator distance for the parallel-hole collimation of Tc-99m was set at 4 mm.

### 2.3. Segmentation and Feature Extraction

Volume of interest (VOI) segmentation was carried out on the open-access LIFEx platform version 7.10 (IMIV/CEA, Orsay, France). The lesions were delineated automatically using the Nestle's adaptive thresholding method (Threshold = $(0.3 \times SUV_{tumor\_mean70\%\_SUVmax}) + SUV_{background\_mean}$). Two experienced nuclear medicine physicians manually inspected the drawn lesions to remove VOIs that were not attributable to the primary tumor, if needed (Figure 1).

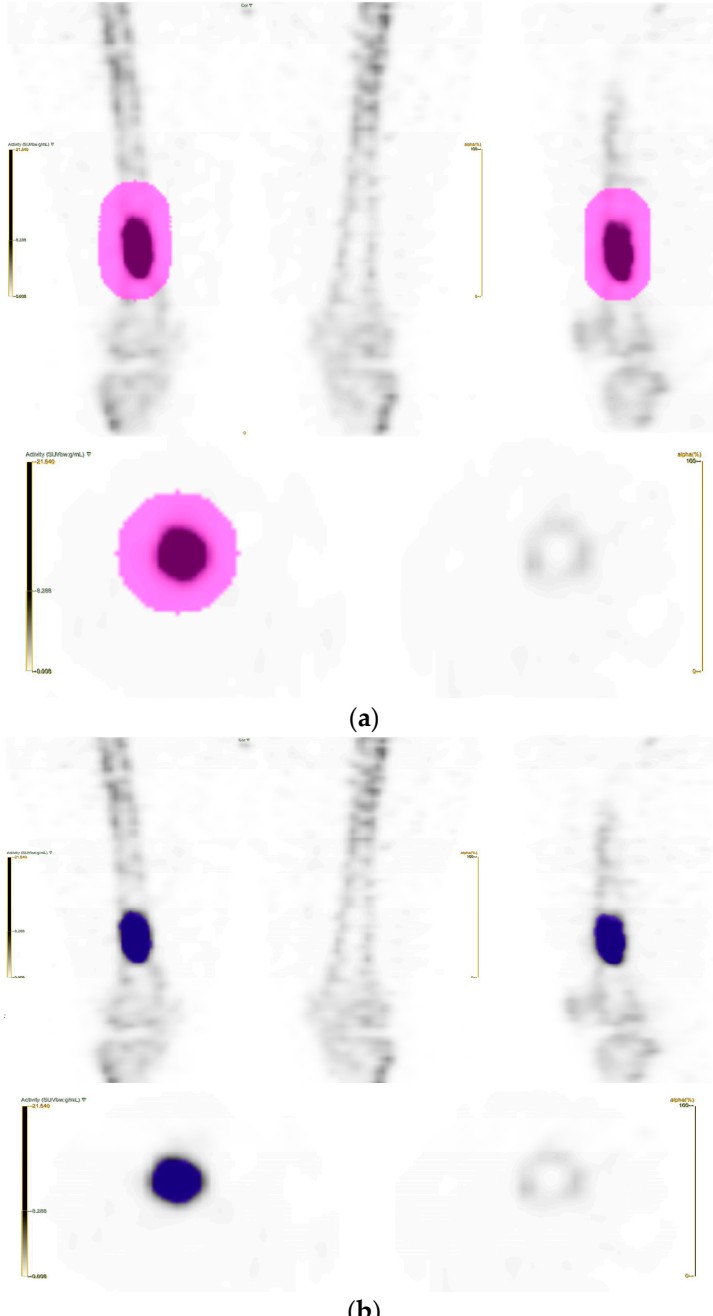

**Figure 1.** (**a**) A large VOI was drawn to include all of the visible tumor. (**b**) With Nestle's adaptive thresholding method, the VOI of the tumor was automatically delineated.

Feature extraction was done on the LIFEx platform. Intensity level numbers were resampled into 64 discrete levels ranging from 0 to 25 of SUV. Forty-two SPECT parameters were extracted and categorized into three main groups: five conventional parameters, six histogram parameters, and thirty-one texture parameters. The thirty-one texture parameters were evaluated using four texture matrices, namely the Gray-Level Co-occurrence Matrix (GLCM), Grey-Level Run Length Matrix (GLRLM), Neighborhood Grey-Level Different Matrix (NGLDM), and Grey-Level Zone Length Matrix (GLZLM), adhering to the most current benchmark values of the Image Biomarkers Standardization Initiative (IBSI). The coefficient of variation (CV), defined as the standard deviation over the mean of SUV distribution, was used instead of the standard deviation in order to satisfy IBSI require-

ments [11]. The full list of extracted radiomics parameters is provided in Supplementary Materials Table S1. All parameters were normalized into Z-values.

### *2.4. Feature Selection and Statistical Analysis*

SPSS software (version 24.0; IBM Corp., Armonk, NY, USA) and R version 3.2.3 (The R Foundation for Statistical Computing, Vienna, Austria) were used for feature selection and statistical analysis.

Feature selection/reduction was performed to reduce the high-dimensional problem from a large number of parameters with co-linearity. With training data, a least absolute shrinkage and selection operator (LASSO) regression was performed with 'glmnet' package in R to select the most important feature related to differentiating ACTs from enchondromas. LASSO regression is commonly used to select relevant features and encourage sparsity in the model [12]. The variables that minimized the mean square error (MSE) were selected for further analysis.

Univariate and multivariate logistic regression analysis for the prediction of diagnosis were performed for selected parameters, as well as classic quantitative parameters, such as SUVmax, SUVmean, and tumor volume. A receiver operating characteristic (ROC) curve analysis was used to dichotomize the selected feature. Significant parameters on multivariate analysis were again tested in the test data.

### 3. Results

### *3.1. Demographics*

A total of 49 patients were included in this retrospective study. A total of 20 (40.8%) patients were diagnosed as grade I chondrosarcoma. Data was randomly split into training (*n* = 32) and test (*n* = 17) data, each with 12 (37.5%) and 6 (35.2%) patients diagnosed as grade I chondrosarcoma, respectively. The patient demographics are summarized in Table 1. The median patient age was 54 years (range 31–45) in training and 49 years (range 19–70) in the test data. All VOIs delineated with Nestle's adaptive thresholding method were considered to be acceptable, and no manual adjustment was made.

**Table 1.** Demographic summary.

|  | Training Data | Test Data |
|---|---|---|
| Patients (*n*) | 32 | 17 |
| Median age (range), years | 54 (31–45) | 49 (19–70) |
| Sex |  |  |
|    Male | 9 (28.1%) | 6 (35.3%) |
|    Female | 23 (71.9%) | 11 (64.7%) |
| Diagnosis |  |  |
|    Enchondroma | 20 (62.5%) | 11 (64.8%) |
|    Grade 1 chondrosarcoma | 12 (37.5%) | 6 (35.2%) |
| Skeletal distribution |  |  |
|    Femur | 16 | 6 |
|    Humerus | 13 | 9 |
|    Tibia | 2 |  |
|    Fibula | 1 | 2 |

### *3.2. Statistical Analysis*

### 3.2.1. Training Data

From training data, LASSO regression revealed that two variables minimize MSE (Figure 2). Two radiomics features, zone-length non-uniformity for zone ($ZLNU_{GLZLM}$) and coarseness for neighborhood grey-level difference ($Coarseness_{NGLDM}$), were found to be the most important features related to differentiating grade I chondrosarcomas from enchondromas. Univariate analysis revealed higher SUVmean, SUVmax, and $ZLNU_{GLZLM}$ values as being significant predictive factors for the diagnosis of grade I chondrosarcoma. Multivariate analysis revealed $ZLNU_{GLZLM}$ to be the only significant independent factor

related with the diagnosis of grade I chondrosarcoma. The results of univariate and multivariate analysis with training data are shown in Table 2.

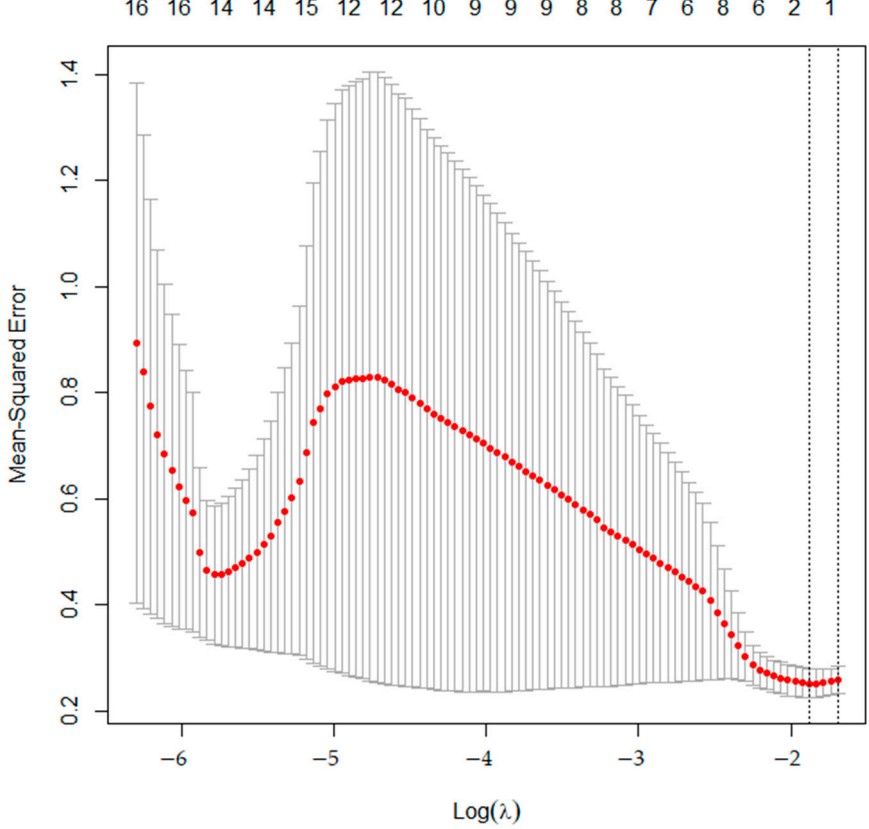

**Figure 2.** The mean squared error (MSE) of training data. MSE (red dots) is minimized when lambda is 2.

**Table 2.** Univariate and Multivariate regression for diagnosis of ACT.

|  | Univariate Analysis | | Multivariate Analysis | |
| --- | --- | --- | --- | --- |
|  | **Coefficient** | *p* | **Coefficient** | *p* |
| SUVmean | −0.35 | 0.052 | | |
| SUVmax | −0.36 | 0.043 | | |
| Volume | −0.29 | 0.107 | | |
| Coarseness$_{NGLDM}$ | 0.29 | 0.112 | | |
| ZLNU$_{GLZLM}$ | −0.38 | 0.032 | −0.38 | 0.032 |

The univariate and multivariate regress areas under the receiver-operating characteristic curve (AUC) for ZLNU$_{GLZLM}$ was 0.721. With a cut-off value of 191.26, the sensitivity and specificity of ZLNU$_{GLZLM}$ was 85.0% and 58.3%, respectively, for differentiating between enchondroma and grade I chondrosarcoma (Figure 3a).

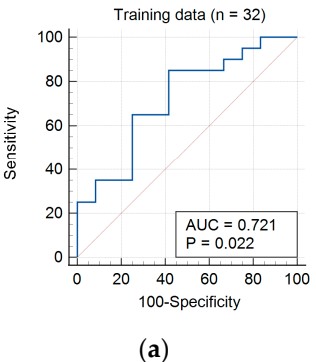
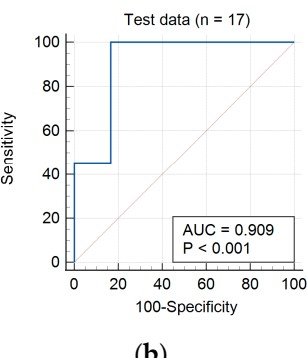

**(a)**                                                      **(b)**

**Figure 3.** AUC (blue lines) of $ZLNU_{GLZLM}$ diagnosis of chondrosarcomas in (**a**) training data and (**b**) test data.

### 3.2.2. Test Data

In test data, higher $ZLNU_{GLZLM}$ was again significantly associated with the diagnosis of grade I chondrosarcoma. The AUC was 0.909. With cutoff of 191.26, the sensitivity and specificity of $ZLNU_{GLZLM}$ was 83.3% and 90.9%, respectively, for differentiating between enchondroma and grade I chondrosarcoma (Figure 3b). The mean values of all extracted parameters in the training data and test data are presented in Supplementary Material Tables S2 and S3.

### 4. Discussion

We evaluated the osteoblastic phenotype of chondroid tumors in HDP SPECT/CT scans via radiomics analysis. By LASSO regression, $ZLNU_{GLZLM}$ and $Coarseness_{NGLDM}$ were selected as the most important factors for differentiation of ACTs from enchondromas in the long bones. Logistic regression has shown $ZLNU_{GLZLM}$ to be the only significant independent factor related with the diagnosis of ACT. The performance of $ZLNU_{GLZLM}$ for differentiating ACTs from enchondromas was also noted in the test cohort.

While certain imaging features have been suggested for the differentiation of enchondromas and chondrosarcomas, distinguishing ACTs from enchondromas remains a challenge even to expert radiologists and pathologists [3,13]. Unlike other tumors, even biopsy does not always provide an accurate result. Currently, the differentiation between ACTs from enchondromas with imaging is an unmet clinical need.

Chondrosarcomas typically demonstrate high radionuclide uptake on bone scintigraphy, compared to enchondromas [14]. SPECT/CT imaging allows for three accurate localizations of uptake when compared with planar scintigraphy. Traditionally, SPECT/CT imaging lacked the ability to provide an accurate quantitative analysis. Recent advances in technology have enabled a feasible semi-quantitative evaluation using SUV, similarly to PET/CT systems [5]. Our previous study demonstrated that ACTs show higher SUVmax compared to enchondromas [6].

There has been increasing interest in the field of radiomics, which have been shown to provide prognostic information for various cancers [15]. However, while radiomics research shows great potential, its current use is mainly confined to the academic literature, and its presently lacking the transition to real-world clinical application. This is partly due to the lack of efficient strategies to develop radiomics into a useful image biomarker guiding clinical decisions [16].

In the field of radiomics, while a large number of parameters are provided, the exact meaning and clinical implications of each value are yet uncertain. The search for one best texture parameter has not yet met consensus.

A large number of features variables within a limited number of data often causes "the curse of dimensionality", leading to overfitting and multiple comparison-related problems [17]. Therefore, feature selection or feature extraction is required to reduce dimensionality. In this study, we applied LASSO regression to perform variable selection [18].

As a result, $\text{ZLNU}_{\text{GLZLM}}$ and $\text{Coarseness}_{\text{NGLDM}}$ were selected for further analysis, and $\text{ZLNU}_{\text{GLZLM}}$ was shown to be the only significant independent factor related with the diagnosis of ACT. While these radiomics features may reflect the tumor heterogeneity similarly to FDG PET/CT [19], the robustness and clinical implications of HDP SPECT/CT radiomics have not been prior evaluated.

Several other studies have investigated the value of MR image-based radiomics in chondroid tumors [9,10,20]. These studies have shown promising results in the discrimination of enchondroma and chondrosarcoma. However, these studies have included chondrosarcoma patients of all grades, and their performance for distinguishing ACTs from enchondromas was not evaluated.

The reproducibility of radiomics is of a major concern [21]. Image segmentation can greatly affect the outcome of the study. Out study aimed to provide consistent segmentation by using Nestle's adaptive thresholding method to automatically define VOI. Different feature extraction methods may provide different values from the same images. To prevent this, the IBSI aims to standardize commonly used radiomics features, and ensure that the same results are used between different extraction software programs [9]. We extracted radiomics features with LIFEx, which complies with ISBI standards.

To our knowledge, this is the first study to evaluate the value of quantitative HDP SPECT/CT radiomics for differentiating ACTs from enchondromas in the long bones.

There are several limitations in our study. This was a retrospective study with a relatively small number of patients. Radiomics studies typically require large amounts of data, and results created from a small number of patients show a high risk of overfitting and model instability [21]. Our results also show signs of instability, where the specificity was increased in the test data compared to training data, while typical well-developed internal validation should show similar, but slightly lower, accuracy for the test data. Also, all cases were obtained from a single center, and only internal validation was performed.

## 5. Conclusions

Radiomics analysis of HDP SPECT/CT images may provide prognostic factor for differentiation of grade I chondrosarcomas from enchondromas, and may have clinical potential.

**Supplementary Materials:** The following supporting information can be downloaded at https://www.mdpi.com/article/10.3390/tomography9050148/s1: Table S1: List of extracted metabolic parameters; Table S2: The mean values of all extracted parameters in the training data; Table S3: The mean values of all extracted parameters in the test data.

**Author Contributions:** Conceptualization: W.H.C. and M.W.J.; methodology: H.Y. and S.H.; validation: H.Y. and W.H.C.; formal analysis: H.Y.; writing—original draft preparation: H.Y.; writing—review and editing; H.Y. and W.H.C.; visualization: H.Y.; supervision: Y.-A.C. All authors have read and agreed to the published version of the manuscript.

**Funding:** This research was supported by a grant from Institute of Information & communications Technology Planning & Evaluation (2021-0-00986).

**Institutional Review Board Statement:** The study was conducted in accordance with the Declaration of Helsinki, and approved by the Institutional Review Board of St. Vincent's Hospital.

**Informed Consent Statement:** The informed consent was waived by the IRB, due to the retrospective nature of the study.

**Data Availability Statement:** The data presented in this study are available on request from the first author. The data are not publicly available due to privacy restrictions on clinical information.

**Conflicts of Interest:** The authors declare no conflict of interest.

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
