# Peer review of "SPECT/CT Radiomics for Differentiating between Enchondroma and Grade I Chondrosarcoma"

_tomography, doi:10.3390/tomography9050148_

Round 1

Reviewer 1 Report

Overall, I believe this is a commendable study. However, there are several areas that I would like to highlight for further clarity and enhancement.

There are no clinical images provided. One of the most crucial aspects of this research appears to be the method of defining the ROI. It would be beneficial for readers if the authors can exemplify which cases allowed for automated ROI extraction and which ones required manual adjustments.

The utilization of the blood pool image is not explained at all. It would be advantageous to provide detailed information on how these images were used in the study.

In the final paragraph of the Methods section, there is mention of "Univariate and multivariate regression analysis." Could you clarify if this refers to logistic regression?

For ease of reading and better organization, it would be helpful to include sub-section titles within the Materials and Methods section.

Author Response

Overall, I believe this is a commendable study. However, there are several areas that I would like to highlight for further clarity and enhancement.

There are no clinical images provided. One of the most crucial aspects of this research appears to be the method of defining the ROI. It would be beneficial for readers if the authors can exemplify which cases allowed for automated ROI extraction and which ones required manual adjustments.

  • Figures were added to describe the VOI delineation process.
  • While our study design was to manually adjust the VOI if needed (this was actually needed in other study involving HNSCC where nearby physiologic activity had to be removed),In this study no images required additional adjustment, probably due to high contrast between tumor and background. We added following phrase: “All VOI delineated with Nestle’ adaptive thresholding method were acceptable, and no manual adjustment was made” to the results section of the manuscript

The utilization of the blood pool image is not explained at all. It would be advantageous to provide detailed information on how these images were used in the study.

  • As retrospective study, the blood pool images were acquired as routine clinical process. While bloodpool images were used in practice for reading (to assess hyperemia), they were not used in this specific study.

In the final paragraph of the Methods section, there is mention of "Univariate and multivariate regression analysis." Could you clarify if this refers to logistic regression?

  • To clarify, we added the word “logistic” in the manuscript.

For ease of reading and better organization, it would be helpful to include sub-section titles within the Materials and Methods section.

  • We added sub-sections in the manuscript as recommended,

Reviewer 2 Report

The Authors have investigated the usefulness of radiomic parameters derived from HDP SPECT/CT in distinguishing atypical cartilaginous tumors (ACTs) from enchondromas. The topic is interesting, considering that distinction between these two entities carries a significant change in patient's management (ACTs need curative surgery; enchondromas benefit of a wait-and-see policy).

The manuscript is overall well written, the study design, although it has been retrospectively conducted (with possible drawbacks of a retrospective analysis), is explained in great details, and results are clearly provided.

However, some minor changes should be made to the text, in my opinion, in order to give higher scientific soundness to the manuscript:

- the authors have found that the texture feature ZLNUGLZLM is the only able to distinguish ACTs (higher values) from enchondromas (lower values), and this is valid for both training and test populations; however, I noticed that specificity of the cut-off value for ZLNUGLZLM was 90.9% in the test population but only 58.3% in the training one: was somehow this difference in sensitivity explored in order to find the reason? The two population share the same proportions in ACTs and enchondromas (35-37% and 62-65% respectively), and the training population is larger than the test population (32 vs 17 patients); however, 72% patients are female in the training population compared with only 65% in the test (maybe osteoporosis which is more frequent in women than in men plays a role in lowering specificity in the training group?). Please explain

-  from the supplementary tables it is evident that another texture feature, the SZEGLZLM, is  significantly higher is ACTs than in enchondromas in both training (P=0.041) and test populations (P=0.031): was this result not confirmed at the univariate and multivariate analyses, so that it has no predictive value? Please explain.

Best regards. 

Author Response

the authors have found that the texture feature ZLNUGLZLM is the only able to distinguish ACTs (higher values) from enchondromas (lower values), and this is valid for both training and test populations; however, I noticed that specificity of the cut-off value for ZLNUGLZLM was 90.9% in the test population but only 58.3% in the training one: was somehow this difference in sensitivity explored in order to find the reason? The two population share the same proportions in ACTs and enchondromas (35-37% and 62-65% respectively), and the training population is larger than the test population (32 vs 17 patients); however, 72% patients are female in the training population compared with only 65% in the test (maybe osteoporosis which is more frequent in women than in men plays a role in lowering specificity in the training group?). Please explain

-> We believe the main cause of varying sensitivity in our study is due to our small sample size, which is our study’s primary weakness, where 1 or 2 of outliers will change the model accuracy drastically, especially in the test group. We have added discussion about the small number of patients in our manuscript.

7% difference in the rate of females in the population is equivalent to 1 patient in the test data, and seems to be insufficient to be the direct cause of this result. However, we do not have information about the incidence of osteoporosis in our study. It will be interesting if information about osteoporosis is gathered, on our further studies.

-  from the supplementary tables it is evident that another texture feature, the SZEGLZLM, is significantly higher is ACTs than in enchondromas in both training (P=0.041) and test populations (P=0.031): was this result not confirmed at the univariate and multivariate analyses, so that it has no predictive value? Please explain.

-> If we tested every variable extracted without feature reduction, we were afraid that we would face multiple comparison problem due to our relatively small sized cohort. Therefore, we had to perform feature reduction with LASSO regression, and only used 2 factors (ZLNUGLZLM and CoarsenessNGLDM) for further evaluation. SZEGLZLM may have predictive value, but our data is insufficient to validate it.

Reviewer 3 Report

Dear authors, 
this a really interesting paper that focuses on a new field of research. Even if particularly well designed, I think that the main limitation of this study is the low sample of patients included. This issue needs to be better underlined in the discussion. As a consequence, the conclusion should be less enthusiastic.
Furthermore, a discussion on the general reproducibility of radiomics analysys would improve the overall quality of the paper.
Lastly, I think that there's a problem with the titles of table 2.

Best regards.

Author Response

this a really interesting paper that focuses on a new field of research. Even if particularly well designed, I think that the main limitation of this study is the low sample of patients included. This issue needs to be better underlined in the discussion. As a consequence, the conclusion should be less enthusiastic.

  • We have added discussion about this study’s weakness due to low sample of patients included, and modified the conclusion to be less enthusiastic

Furthermore, a discussion on the general reproducibility of radiomics analysys would improve the overall quality of the paper.

  • We have added a discussion on the general reproducibility of radiomics in the manuscript.

Lastly, I think that there's a problem with the titles of table 2.

  • We have fixed the titles of table 2 in the manuscript.